# Predictors of Depressive Symptoms in Caregivers of Children with Attention-Deficit/Hyperactivity Disorder: A One-Year Follow-Up Study

**DOI:** 10.3390/ijerph18168835

**Published:** 2021-08-22

**Authors:** Wen-Jiun Chou, Ray C. Hsiao, Chih-Cheng Chang, Cheng-Fang Yen

**Affiliations:** 1Department of Child and Adolescent Psychiatry, Chang Gung Memorial Hospital, Kaohsiung Medical Center, Kaohsiung 83301, Taiwan; wjchou@cgmh.org.tw; 2College of Medicine, Chang Gung University, Taoyuan 33302, Taiwan; 3Department of Psychiatry and Behavioral Sciences, University of Washington School of Medicine, Seattle, WA 98195, USA; rhsiao@u.washington.edu; 4Department of Psychiatry, Children’s Hospital and Regional Medical Center, Seattle, WA 98105, USA; 5Department of Psychiatry, Chi Mei Medical Center, Tainan 70246, Taiwan; 6Department of Health Psychology, College of Health Sciences, Chang Jung Christian University, Tainan 71101, Taiwan; 7Department of Psychiatry, Kaohsiung Medical University Hospital, Kaohsiung 80708, Taiwan; 8Department of Psychiatry, School of Medicine, Kaohsiung Medical University, Kaohsiung 80708, Taiwan; 9College of Professional Studies, National Pingtung University of Science and Technology, Pingtung 91201, Taiwan

**Keywords:** attention-deficit/hyperactivity disorder, behavioral problem, caregiver, depression, family support, predictor

## Abstract

This 1-year follow-up study examined the predictive values of the demographics, depressive symptoms, stress-coping orientations, and perceived family support of caregivers as well as the internalizing, externalizing and ADHD symptoms of children with attention-deficit/hyperactivity disorder (ADHD) at baseline on the depressive symptoms of the caregivers after 1 year. A total of four hundred caregivers of children with ADHD were recruited. The baseline levels of the caregivers’ depressive symptoms, stress-coping orientations, and perceived family support and the internalizing and externalizing problems of the children were assessed using the Center for Epidemiological Studies Depression Scale, the Coping Orientation to Problems Experienced, Family Adaptation, Partnership, Growth, Affection, Resolve Index, and the Child Behavior Checklist For Ages 6–18, respectively. Their predictions for the caregiver’s depressive symptoms 1 year after the baseline were examined using linear regression analysis. In total, 382 caregivers of children with ADHD underwent the follow-up assessment 1 year from the baseline. A marital status of being separated or divorced, less effective coping and depressive symptoms orientation, and children with internalizing problems and ADHD symptoms at baseline were positively associated with the caregivers’ depressive symptoms at follow-up, whereas the caregivers’ perceived family support and an emotion-focused coping orientation at baseline were negatively associated with depressive symptoms at follow-up. Multiple characteristics of the caregivers and children with ADHD at baseline predicted the caregivers’ depressive symptoms 1 year later.

## 1. Introduction

### 1.1. Depression in Caregivers of Children with Attention-Deficit/Hyperactivity Disorder

Depression is prevalent among caregivers of children with attention-deficit/hyperactivity disorder (ADHD) [1,2,3,4]. The negative effects of caregiver depression on children with ADHD have been well established. Follow-up studies have revealed that caregiver depression predicts the exacerbation or persistence of ADHD symptoms [5], conduct disorder symptoms [6,7], and oppositional defiant disorder symptoms [8] in children with ADHD. In addition, studies have reported an association of caregiver depression with dysfunctional parenting skills (e.g., imposing corporal punishment) [9], nonoptimal reinforcement [10], lack of or overly strict disciplining [11,12], and low sensitivity and warmth during interactions with children with ADHD [11,13,14]. Moreover, studies have found that targeting parental depression may enhance the effects of behavioral interventions on the parenting skills of caregivers and the oppositional and noncompliant behaviors of children [15]. Therefore, screening for depressive symptoms and administering interventions targeting depression among the caregivers of children with ADHD are crucial in clinical practice [6,16].

### 1.2. Factors Predicting Caregivers’ Depressive Symptoms

The determination of factors that predict caregiver depressive symptoms is essential for developing prevention and intervention programs for the caregivers of children with ADHD. Previous studies have demonstrated the cross-sectional association of caregiver depression with several caregiver- and child-related factors. Caregiver factors include being female [16], low income [4], not living with their partner [17], and being the only caregiver in the family [4]. Child factors include ADHD symptoms [16], hyperactive/impulsive, and combined types of ADHD were significant predictors of depression [4], internalizing problems such as depression and anxiety [18], and externalizing problems [19]. However, no prospective study has examined the predictors of depressive symptoms in the caregivers of children with ADHD.

### 1.3. Prediction for Depressive Symptoms in Caregivers Based on Their Stress-Coping Orientations

The caregivers of children with ADHD experience considerable parenting stress [20] and courtesy stigma [21]. Stress-coping strategies adopted by caregivers to cope with stress and the effectiveness of these strategies may affect the mental health of caregivers. Problem-solving, emotion-focused, and less effective strategies are the three most commonly used stress-coping orientations [22,23,24,25]. Problem-focused coping constitutes actively modifying or eliminating sources of stress to manage stressful situations [22,23]. Emotion-focused coping involves exerting regulative efforts to diminish the emotional consequences of stressful events [22,23]. Less effective coping strategies include behavioral and mental disengagement to avoid directly dealing with stressful situations [23]. A cross-sectional study including the mothers of children with intellectual disabilities reported that the adoption of problem-coping strategies was associated with higher positive affect, whereas active-avoidance coping was associated with higher negative affect and increased anxiety and depression [26]. However, no prospective study has examined whether various stress-coping orientations can predict depressive symptoms in the caregivers of children with ADHD. If stress-coping orientations can predict depressive symptoms in caregivers, then effective problem-solving strategies should be included in programs designed for improving caregiver mental health.

### 1.4. Prediction for Depressive Symptoms in Caregivers Based on Internalizing and Externalizing Problems of Children with ADHD

Internalizing (e.g., depressive, anxiety, and somatic symptoms) and externalizing (e.g., oppositional defiance and conduct problems) symptoms are prevalent in children with ADHD [27,28,29,30,31,32]. A prospective study found that the parents of children with ADHD and internalizing and externalizing comorbidities reported poorer family quality of life than the parents of children with ADHD alone did [33]. Additional prospective studies examining whether the internalizing and externalizing symptoms of children with ADHD can predict caregiver depressive symptoms may provide evidence for the effects of caregiver depressive symptoms on the comorbid behavioral problems of children with ADHD.

### 1.5. Prediction for Depressive Symptoms in Caregivers Based on Perceived Family Support and Marriage Status

Family support is a microsystemic factor [34] that may help caregivers of children with ADHD maintain their mental health [35,36,37]. Additional prospective studies should examine whether the protective effects of family support still exist when other microsystemic factors (e.g., children’s internalizing and externalizing problems) and individual factors (e.g., caregiver stress-coping orientations) are considered together. Research has found that parents of children with ADHD are more likely to divorce compared to those of children without ADHD [38], and divorce can also cause health-related immune alterations and increase the risk of depressive problems [39]. Whether a marriage status of separation or divorce can predict caregivers’ depressive problems warrants further study.

### 1.6. Persistent Depressive Symptoms in Caregivers of Children with ADHD

Research has found that caregivers of children with ADHD had heavy care burdens [40,41,42] and high parenting stress [43,44]. Caregivers might also experience public stigma toward ADHD [45] and have affiliate stigma toward themselves, which may further increase their psychological distress [46] and the risk of depressive problems [47]. It is reasonable to hypothesize that caregiver depressive symptoms may persist or even worsen during the growth stages of children with ADHD; therefore, depressive symptoms at baseline may predict the non-remitted depressive symptoms at follow-up. However, whether this predictive effect exists warrants further study.

### 1.7. Aims of This Study

This prospective study followed the participants of the Study on Affiliate Stigma in Caregivers of Children with ADHD [47] 1 year after the initial study. In the previous analysis, we found that the affiliate stigma that the caregivers feel towards themselves at baseline positively predicted their children’s affective and somatic problems 1 year later, after controlling for the caregivers’ depressive symptoms and children’s behavioral problems at baseline [48]. In the present study, we examined the predictive values of the demographics (gender, age, education level, and marriage status), depressive symptoms, stress-coping orientations and perceived family support of caregivers and the demographics (gender and age), ADHD symptoms, and internalizing and externalizing problems of children with ADHD at baseline the depressive symptoms of the caregivers after 1 year. We hypothesized that the demographics, depressive symptoms, perceived family support, and various stress-coping orientations of the caregivers at baseline could predict the caregivers’ depressive symptoms at follow-up. Moreover, ADHD symptoms and internalizing and externalizing problems of children with ADHD at baseline could predict caregivers’ depressive symptoms at follow-up.

## 2. Methods

### 2.1. Participants and Procedures

The procedure used for recruiting participants at baseline in the Study on Affiliated Stigma in Caregivers of Children with ADHD was described elsewhere [47]. In brief, 400 caregivers of children aged between 6 and 18 years diagnosed as having ADHD according to the criteria of the *Diagnostic and Statistical Manual of Mental Disorders, Fifth Edition* [49] were recruited from the child and adolescent psychiatric outpatient clinics of two medical centers in Kaohsiung, Taiwan. Caregivers’ depression, stress-coping orientations, and perceived family support as well as their children’s ADHD and internalizing and externalizing problems were assessed. These caregivers of children with ADHD were contacted at outpatient clinics 1 year later and were invited to undergo the follow-up assessment.

The Institutional Review Boards of Kaohsiung Chang Gung Memorial Hospital (approval number: 202000605A3; date of approval: 15 May 2020) and Kaohsiung Medical University Hospital (approval number: KMUHIRB-E(I)-20200111; date of approval: 1 June 2020) approved the study. All participants provided written informed consent before undergoing the assessment.

### 2.2. Measures

#### 2.2.1. Depressive Symptoms

The 20-item Mandarin Chinese version of the Center for Epidemiological Study-Depression Scale (CES-D) was used to assess the severity of depressive symptoms in the caregivers at baseline and at follow-up [50,51]. Caregivers were asked how often they experienced each depressive symptom in the preceding month. The response categories were (0) rarely or none of the time (less than 1 day), (1) some or a little of the time (l–2 days), (2) occasionally or a moderate amount of the time (3–4 days), or (3) most or all of the time (5–7 days). A higher total score indicates more severe depressive symptoms. Cronbach’s α for the scale in the present study was 0.88. A CES-D cutoff of ≥21 was used to identify individuals with a high level of depressive symptom [52].

#### 2.2.2. Stress-Coping Orientations

The 53-item Coping Orientation to Problems Experienced (COPE) evaluation was used to assess the caregivers’ stress-coping orientations at baseline. The COPE measured three domains of coping orientations: problem-focused coping, emotion-focused coping, and less effective coping [23,53]. Every item was rated on a 4-point Likert scale ranging from 1 (*usually don’t do this at all*) to 4 (*usually do this a lot*). A higher total score for a particular domain indicates that participants are more likely to cope with stress by using that strategy. Cronbach’s α for the three domains in the scale in the present study ranged from 0.68 to 0.78.

#### 2.2.3. Perceived Family Support

The five-item Mandarin Chinese version of the Family Adaptation, Partnership, Growth, Affection, Resolve (APGAR) Index was used to examine the caregivers’ perceived family support at baseline [54,55]. Every item was rated on a 4-point Likert scale ranging from 1 (*never*) to 4 (*always*). A higher total score represents a higher level of perceived family support. Cronbach’s α for the scale in the present study was 0.82.

#### 2.2.4. Children’s Internalizing and Externalizing Problems and ADHD Symptoms

The caregiver-reported Chinese Version of the Child Behavior Checklist for Ages 6–18 (CBCL/6-18) was used to measure the children’s behavioral problems at baseline [56,57]. We used the recommended T-score transformations of the raw behavior scores that were adjusted for age and sex differences in behavior found in normative samples. We used the domains of internalizing and externalizing problems and ADHD symptoms for analysis. Because the CBCL/6-18 defined those with a T score higher than 70 (two standard deviation of the mean) as having severe symptoms [56,57], we used a cutoff of >70 to identify children with severe symptoms.

#### 2.2.5. Demographic Characteristics

The caregivers’ sex, age, and marital status, and the children’s sex, and age were collected at baseline. We also determined the caregivers’ total length of education by calculating their total years of receiving formal education.

### 2.3. Statistical Analysis

Data analysis was performed using SPSS 24.0 (SPSS, Chicago, IL, USA). The variables examined in the present study were presented as percentages, means, and standard deviations (SD). We used skewness and kurtosis to examine whether the continuous variables were normally distributed. According to Kim [58], the absolute skewness values < 2 and kurtosis values < 7 indicate normal distribution if the sample size was larger than 300. The predictive effects of the caregivers’ demographic characteristics, depressive symptoms at baseline, perceived family support, and orientations of stress-coping strategies and the children’s demographic characteristics and behavioral problems and ADHD symptoms (independent variables) on the caregivers’ depressive symptoms at follow-up (dependent variable) were first examined using univariate linear regression analysis. Predictors that were significantly associated with depressive symptoms in univariate linear regression further entered into a backward selection of multivariate linear regression analysis. A two-tailed *p* value of <0.05 was used to present statistical significance. We used tolerance, variance inflation factor (VIF), and a condition index to examine the multicollinearity of the multiple regression analysis. A value of ≤0.1 for tolerance, >2.5 for VIF, and >30 for the condition index indicate concern for multicollinearity among variables [59]. The Durbin Watson statistic was used test the autocorrelation in the residuals.

## 3. Results

In total, 382 (95.5%, 308 female and 74 male) caregivers of children with ADHD underwent the follow-up assessment 1 year later. No difference in the caregivers’ sex (χ^2^ = 0.766, *p* = 0.381), marital status (χ^2^ = 1.998, *p* = 0.158), age (*t* = 0.475, *p* = 0.635), educational level (*t* = −1.325, *p* = 0.186), CES-D scores (*t* = −0.271, *p* = 0.787), perceived family support (*t* = −0.384, *p* = 0.701), or stress-coping orientations (*t* = −1.214 to −0.204, *p* = 0.225–0.838) nor any difference in the children’s sex (χ^2^ = 0.845, *p* = 0.358), age (*t* = −0.903, *p* = 0.367), or behavioral problems (*t* = −1.121 to −0.672, *p* = 0.263–0.502) were found between caregivers who participated at baseline and who did not undergo the follow-up assessment.

Table 1 lists the caregivers’ depressive symptoms at baseline and follow-up; the caregivers’ demographic characteristics, perceived family support, and stress-coping orientations at baseline; and the children’s demographic characteristics, behavioral, and ADHD problems at baseline. In total, 83 (21.7%) and 89 (23.3%) caregivers reported a high level of depressive symptoms at baseline and at-follow-up, respectively. Of the 83 caregivers with a high level of depressive symptoms at baseline, 55 (66.3%) still had a high level of depressive symptoms at follow-up. Of the 299 caregivers with a low level of depressive symptoms at baseline, 34 (11.4%) turned to report a high level of depressive symptoms at follow-up. Based on the scores of the CBCL/6-18, the proportions of children with severe internalizing, externalizing, and ADHD symptoms were 13.1%, 17.3%, and 21.7%, respectively. The absolute skewness values of the continuous variables ranged from 0.101 to 1.034; the absolute kurtosis values ranged from 0.014 to 1.170. The skewness and kurtosis values indicated that the continuous variables were normally distributed.

Table 2 presents the results of the univariate linear regression examining the associations of the caregivers’ and children’s factors at baseline with the caregivers’ depressive symptoms at follow-up. Regarding the caregiver factors, the results indicate that compared to the caregivers who were married or who cohabited at baseline, the caregivers who were separated or divorced had greater depressive symptoms at follow-up. Depressive symptoms at baseline and the orientation of less effective coping were positively associated with depressive symptoms at follow-up, whereas perceived family support and the problem-focused coping and emotion-focused coping orientations were negatively associated with depressive symptoms at follow-up. Regarding child factors, internalizing and externalizing problems and ADHD symptoms were positively associated with the caregivers’ depressive symptoms at follow-up.

Factors that were significantly associated with the caregivers’ depressive symptoms in univariate linear regression analysis were selected into a backward selection of multivariate linear regression analysis to examine their association with the caregivers’ depressive symptoms at follow-up (Table 3). The results of Model 1 demonstrated that a marital status of being separated or divorced and a less effective coping orientation as well as the children with internalizing problems and ADHD symptoms at baseline were positively associated with the caregivers’ depressive symptoms at follow-up, whereas the caregivers’ perceived family support and an emotion-focused coping orientation at baseline were negatively associated with depressive symptoms at follow-up. The caregivers’ depressive symptoms at baseline were further selected into multivariate linear regression analysis. The results of Model 2 indicate that depressive symptoms at baseline were significantly associated with depressive symptoms at follow-up in the caregivers. Multicollinearity among the variables was not detected because the tolerance and VIF values ranged from 0.758 to 0.988 and from 1.012 to 1.319, respectively, and the condition index value was 26.976. The value of the Durbin Watson statistic was 1.974, indicating a weak autocorrelation in the residuals.

## 4. Discussion

The results of this study indicate that over one-fifth of the caregivers had a high level of depressive symptoms. Multiple characteristics of the caregivers, namely marital status, perceived family support, emotion-focused and less effective coping strategy orientations, and depressive symptoms at baseline and characteristics of children with ADHD, namely internalizing problems and ADHD symptoms at baseline predicted the caregivers’ depressive symptoms 1 year later. The results indicated that the mental health of caregivers can be affected by both individual and environmental factors.

### 4.1. Caregivers’ Depressive Symptoms

The results of the present study reveal that 21.7% and 23.3% of the caregivers reported a high level of depressive symptoms at baseline and at follow-up, respectively. Moreover, nearly two-thirds (66.3%) of caregivers reported non-remitted depressive symptoms in the 1-year follow-up period. Depressive symptoms at baseline had the most powerful predictive effect on depressive symptoms at follow-up. Parental depressive disorders increase the risk of offspring ADHD [18,60,61]; the high comorbidity of parental depressive disorders and child ADHD can be the biological result of familial aggregation. Because of the negative effects of depressive problems on the health of caregivers and children with ADHD, depressive symptoms in the caregivers of children with ADHD warrants early identification and intervention. Caregivers with depressive symptoms require individual treatment or expanded parenting intervention targeting their cognition [62]. Intervention programs designed to alleviate depressive problems that caregivers may face can also be beneficial for the children who these caregivers take care of [63].

### 4.2. Caregivers’ Perceived Family Support and Marital Status

The results of the present study show that the caregivers’ perception of having low family support at baseline predicted depressive symptoms 1 year later. Congruent with the results of previous studies [35,36,37], the results of this study also supported that family support is a factor [34] that may help caregivers of children with ADHD maintain their mental health. The present study also found that a marital status of being separated or divorced at baseline predicted depressive symptoms 1 year later. Divorce can not only cause health-related immune alterations and increase the risk of depression [39] but can also result in a lack of a coadjutant to manage the behaviors of children with ADHD, thus increasing the caregivers’ parental distress. A previous study on the parents of children with psychopathology found that parents who did not live together had a higher level of depression than did those who lived together [17]. The results of this study indicated that enhancing family support, especially for caregivers who are separated or divorced, is important to improve caregiver mood regulation.

### 4.3. Caregivers’ Stress-Coping Orientations

Coping refers to the process of managing psychological stress through cognitive or behavioral efforts [64]. The results of the present study showed that the caregivers’ adoption of less effective coping mechanisms at baseline increased the risk of depressive symptoms at follow-up. Less effective coping strategies measured in the COPE include behavioral and mental disengagement [23]. Behavioral disengagement is a coping strategy in which an individual reduces their effort to deal with a stressor [23]. The caregivers of children with ADHD may experience the difficult position of being unable to effectively change their children’s behaviors despite all efforts; this may result in learned helplessness [19]. People in a state of helplessness tend to adopt behavioral disengagement as a strategy to cope with stress [23]. Mental disengagement is a coping strategy in which individuals attempt to distract themselves from thinking about the behavioral dimension or goal with which the stressor is interfering [23]. Caregivers who adopt mental disengagement as a coping strategy may engage in alternative activities to take their minds off parenting difficulties or escape through sleep or by spending too much time watching television or using the Internet. Adoption of less effective coping strategies such as behavioral and mental disengagement often impedes the development of adaptive coping [65] and thus increases the risk of psychological distress and depression [66]. This study demonstrated that caregivers with emotion-focused coping orientations at baseline were negatively associated with depressive symptoms at follow-up. Parents may experience multiple sources of stress when taking care of children with ADHD such as managing their children’s behaviors, supervising the completion of their children’s daily routines, maintaining relationships among family members, and craving social support [35]. Emotion-focused coping such as seeking socioemotional support, positive reinterpretation, acceptance of stressors, and perceived growth in the process of managing stress may diminish the negative emotional consequences of stressful events [22,23]. The results of this study indicate that mental health professionals must help the caregivers of children with ADHD to develop effective stress-coping strategies but not avoidance or disengagement strategies to reduce the risk of depression.

### 4.4. Children’s ADHD Symptoms and Internalizing Problems

The proportions of children with severe internalizing, externalizing, and ADHD symptoms determined by the cutoff of a T score > 70 (2 SD) on the CBCL/6-18 were 13.1%, 17.3% and 21.7%, respectively, in this study. Because only about 2.3% of the children in the general population have a T score > 70 on the CBCL/6-18 [56,57,67], the proportions of children with severe internalizing, externalizing, and ADHD symptoms in this study were high. Congruent with our hypothesis and the results of a cross-sectional study [16], our study results demonstrate that the level of children’s ADHD symptoms at baseline increased the risk of depressive symptoms in caregivers 1 year later. ADHD symptoms significantly relate to impairment in peer, child–parent, and student–teacher relationships [68,69,70,71]. Moreover, children with ADHD are prone to accidental injury and trauma [72,73] and have higher mortality [74] than those without ADHD. All of the adverse consequences of ADHD increase the caregivers’ burden of care and psychological distress and may contribute to their depression. Previous studies have revealed that the treatment of children with ADHD exerts a favorable effect on their caregivers’ depressive symptoms [75,76].

A cross-sectional study revealed that the total behavioral problems of children with ADHD, as determined using the CBCL, were related to caregiver depressive symptoms [19]. However, the findings of the present study indicate that children’s internalizing but not externalizing problems at baseline predicted caregiver depressive symptoms 1 year later. ADHD and internalizing problems, such as anxiety and depression, are correlated in families [18]. The prediction of the caregivers’ depressive symptoms based on their children’s internalizing problems may be the result of familial aggregation. Moreover, compared to individuals with ADHD but no significant internalizing problems, those with comorbid ADHD and significant internalizing problems had poorer neurocognitive function [77], poorer psychosocial functioning [78,79], and a lower response rate to stimulant treatment [80]. These adverse consequences of comorbid internalizing problems may increase caregiver stress and worsen depressive symptoms.

### 4.5. Strengths and Limitations

This is one of the first prospective studies to examine the predictive values of stress-coping orientations and perceived family support in caregivers and the behavioral problems of children with ADHD on the depressive symptoms of caregivers of children with ADHD. The present study identified several modifiable predictors of caregiver depressive symptoms and provided evidence to develop prevention programs for reducing the risk of depressive symptoms in caregivers of children with ADHD. However, the present study has several limitations. First, this study only collected reports from caregivers. Common-method variance might occur due to the use of a single data source. Second, whether caregivers received pharmacological or psychological treatment for their depressive symptoms was not determined in this study. Treatment might affect the association between potential predictors at baseline and depressive symptoms at follow-up. We also did not collect the data regarding the treatment for the children’s ADHD at follow-up. Moreover, we did not determine the history of depressive disorders and treatment in caregivers with depressive symptoms. A long history of depressive disorders and a lack of treatment may have deleterious effects on caregiver–child interaction and the recovery of the caregivers’ depressive symptoms. Third, this study examined the predictive values of caregiver stress-coping orientations and perceived family support and the effect of children’s internalizing and externalizing problems on the depressive symptoms of caregivers. The predictive values of other individual and environmental factors warrant further study.

## 5. Conclusions

The present study results indicated that mental health can be affected by both individual and environmental factors among the caregivers of children with ADHD. These predictors should be considered while developing prevention and intervention programs for the depressive symptoms that may be experienced by caregivers. Because the depressive symptoms experienced by caregivers may not remit spontaneously, mental health professionals should routinely survey and provide necessary treatment for depressive symptoms experienced by caregivers. Helping caregivers to develop effective stress-coping strategies and treating children’s ADHD and internalizing behavioral problems may promote caregiver mental health. In addition, mental health professionals should enhance psychosocial support for the caregivers of children with ADHD to prevent the development of depressive symptoms.

## Figures and Tables

**Table 1 ijerph-18-08835-t001:** Caregivers’ demographics, depressive symptoms, perceived family support, stress-coping orientation and children’s demographics, behavioral, and ADHD symptoms (*n* = 382).

Variable	*n* (%)	Mean (SD)	Range
*Caregivers’ characteristics*			
Depressive symptoms at baseline		14.4 (9.9)	0–55
Level of depressive symptoms at baseline			
Low	299 (78.3)		
High	83 (21.7)		
Depressive symptoms at follow-up		13.8 (9.6)	0–45
Level of depressive symptoms at follow-up			
Low	293 (76.7)		
High	89 (23.3)		
Gender			
Female	308 (80.6)		
Male	74 (19.4)		
Age (years)		43.1 (7.0)	23–69
Length of education (years)		14.2 (3.2)	0–28
Marriage status			
Married or cohabited	307 (80.4)		
Separated or divorced	75 (19.6)		
Perceived family support		15.8 (3.3)	6–20
Problem-focused coping orientation		62.0 (11.4)	20–80
Emotion-focused coping orientation		50.9 (8.3)	20–77
Less effective coping orientation		23.6 (5.6)	12–44
*Children’s characteristics*			
Gender			
Girl	76 (19.9)		
Boy	306 (80.1)		
Age (years)		10.9 (3.2)	6–18
Internalizing problems		60.3 (10.3)	33–87
Severe internalizing problems			
No	332 (86.9)		
Yes	50 (13.1)		
Externalizing problems		59.8 (10.5)	33–84
Severe Externalizing problems			
No	316 (82.7)		
Yes	66 (17.3)		
ADHD symptoms		64.1 (7.7)	50–80
Severe ADHD symptoms			
No	299 (78.3)		
Yes	83 (21.7)		

**Table 2 ijerph-18-08835-t002:** Predictors of depressive symptoms at follow-up in caregivers of children with ADHD: univariate regression analysis.

Variable	Depressive Symptoms at Follow-Up	
	B (SE)	95% CI
*Caregiver*		
Male ^a^	−2.182 (1.277)	−4.693–0.328
Age	−0.070 (0.072)	−0.212–0.072
Education	−0.206 (0.160)	−0.521–0.109
Marriage status of separation or divorce ^b^	3.142 (1.265) *	0.654–5.630
Depressive symptoms at baseline	0.714 (0.038) ***	0.638–0.789
Perceived family support	−1.008 (0.146) ***	−1.296–−0.720
Problem-focused coping	−0.158 (0.044) ***	−0.244–−0.072
Emotion-focused coping	−0.127 (0.060) *	−0.246–−0.008
Less effective coping	0.623 (0.086) ***	0.455–0.791
*Children’s characteristics*		
Boy ^c^	0.378 (1.269)	−2.117–2.874
Age	−0.149 (0.157)	−0.459–0.160
Internalizing problems	0.348 (0.046) ***	0.258–0.438
Externalizing problems	0.260 (0.047) ***	0.169–0.352
ADHD symptoms	0.356 (0.063) ***	0.232–0.480

ADHD: attention-deficit/hyperactivity. ^a^ Female as the reference; ^b^ marriage status of being married or cohabited as the reference; ^c^ girl child as the reference. *: *p* < 0.05; *** *p* < 0.001.

**Table 3 ijerph-18-08835-t003:** Predictors of depressive symptoms at follow-up: multivariate regression analysis.

Variable	Depressive Symptoms at Follow-Up			
	Model 1		Model 2	
	B (SE)	95% CI	B (SE)	95% CI
Caregivers’ marital status of separation or divorce ^a^	2.215 (1.092) *	0.067–4.364	2.045 (0.899) *	0.988–1.012
Caregivers’ perceived family support	−0.539 (0.145) ***	−0.823–−0.255		
Caregivers’ emotion-focused coping	−0.180 (0.057) **	−0.292–−0.067		
Caregivers’ less effective coping	0.549 (0.086) ***	0.380–0.717	0.144 (0.070) *	0.833–1.200
Children’s internalizing problems	0.192 (0.047) ***	0.100–0.284	0.088 (0.040) *	0.758–1.319
Children’s ADHD symptoms	0.138 (0.061) *	0.067–4.364	0.120 (0.050) *	0.831–1.204
Caregivers’ depressive symptoms at baseline			0.621 (0.042) ***	0.779–1.284

ADHD: attention-deficit/hyperactivity. ^a^ Marriage status of being married or cohabited as the reference. *: *p* < 0.05; **: *p* < 0.01; ***: *p* < 0.001.

## Data Availability

The data will be available upon reasonable request to the corresponding authors.

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
