# Peer review of "Predictors of Depressive Symptoms in Caregivers of Children with Attention-Deficit/Hyperactivity Disorder: A One-Year Follow-Up Study"

_ijerph, 2021, doi:10.3390/ijerph18168835_

Round 1

Reviewer 1 Report

It’s an interesting paper but it can be improved with some minor language revisions and adding more information in introduction section and link it better with the results.

In result section, the authors produce results that have not been introduced in the beginning. The paper will be improved if they do. 

Author Response

Comment
It’s an interesting paper but it can be improved with some minor language revisions and adding more information in introduction section and link it better with the results. In result section, the authors produce results that have not been introduced in the beginning. The paper will be improved if they do.

Response

Thank you for your comments. As discussed below, we have revised our manuscript based on your comments. Please let us know if we need to provide anything else regarding this revision.

  1. We found that we did not introduce the predictions of depressive symptoms and marriage status at baseline for depressive symptoms at follow-up in caregivers of children with ADHD in Introduction section. We added two paragraphs into Introduction section as below to describe them.

“1.5. Prediction for significant depressive symptoms in caregivers based on perceived family support and marriage status

Research has found that parents of children with ADHD are more likely to divorce compared with those of children without ADHD [38], as well as that divorce can cause health-related immune alterations and increase the risk of depressive problems [39]. Whether a marriage status of separation or divorce can predict caregivers’ depressive problems warrants further study. Please refer to line 104-109.

“1.6.Persistent depressive symptoms in caregivers of children with ADHD

Research has found that caregivers of children with ADHD had heavy care burdens [40-42] and high parenting stress [43,44]. Caregivers might also experience public stigma toward ADHD [45] and have affiliate stigma toward themselves, which may further increases their psychological distress [46] and the risk of depressive problems [47]. It is reasonable to hypothesize that caregivers’ depressive symptoms may persist or even worsen during the growth stage of children with ADHD; therefore, significant depressive symptoms at baseline may predict the non-remitted depressive symptoms at follow-up. However, whether the predictive effect exists warrants further study.” Please refer to line 110-118.

  1. We revised the section “1.7. Aims of this study” as below to include the variables examined in this study. Please refer to line 119-134.

“1.7. Aims of this study

This prospective study followed the participants of the Study on Affiliate Stigma in Caregivers of Children with ADHD [47] 1 year later. In the previous analysis, we found that caregivers’ affiliate stigma at baseline positively predicted children’s affective and somatic problems 1 year later after controlling for caregivers’ depression and children’s behavioral problems at baseline [48]. In the present study we examined the predictive values of demographics (gender, age, education level, and marriage status), depressive symptoms, stress-coping orientations and perceived family support of caregivers, and demographics (gender and age), ADHD symptoms, and internalizing and externalizing problems of children with ADHD at baseline for caregivers’ significant depressive symptoms after 1 year. We hypothesized that demographics, significant depressive symptoms, perceived family support, and various stress-coping orientations of caregivers at baseline could predict caregivers’ significant depressive symptoms at follow-up. Moreover, ADHD symptoms, internalizing and externalizing problems of children with ADHD at baseline could predict caregivers’ significant depressive symptoms at follow-up.

  1. We also revised the wording thorough the manuscript. We invited a native English-editor to edit the writing.

Reviewer 2 Report

This longitudinal study investigated predictive values of the stress-coping orientations and perceived family support of caregivers and the internalizing and externalizing problems of children with ADHD.

Intro: well-written, no comments here

Methods

  • were all values normally distributed?
  • what kind of stepwise selection was used (backward, forward?)
  • how was education level determined?
  • what statistical software was used?

Results

  • Table 1. Was the highest value for years of education truly 28 years?
  • Table 1. Please add n and % of common ADHD comorbidities
  • Table 2. Please report 95% CI for comparison of metric values
  • Table 3. Please provide the reference category for the independent categorical variables (e.g. marriage status).
  • Table 3. What was the Durbin Watson statistic?
  • Please also describe the cohort in more detail. Instead of just reporting the mean values, you might consider to report number of people with different degrees of depression: The CES-D also provides cutoff scores (e.g., 16 or greater) that aid in identifying individuals with depression.
  • Please report use of antidepressive medication in the participants.
  • Please detail the ADHD symptoms.
  • Consider to report how your cohort is compared to other cohorts; e.g., normative data are available for the Child Behavior Checklist For Ages.
  • Instead of using significant variables listed in Table 2 for stepwise multiple regression analysis, you can use all variables with backward selection and AIC criterion.
  • Was there a correlation between different coping styles and depression?

Discussion

  • "21.7% of caregivers were 223 identified to have significant depression" this belongs to the results.
  • Coping: Please also discuss your findings in terms of the categorization into active and passive coping.

The main problem in this paper is the missing information about relevant cofactors: duration of depression (existence of clinical significant depression according to DSM criteria), changes of therapy of ADHD symptoms, changes of ADHD symptoms during study period. In general the term "depression" is somehow misleading, because the CES-D asks symptoms associated with depression, but it is not a instrument to diagnose depression. The paper would also profit from more detailed descriptive statistics, e.g.  the number of different depression symptoms according to the CES-D, different ADHD symptoms.

Author Response

Thank you for your comments. As discussed below, we have revised our manuscript based on your comments. Please let us know if we need to provide anything else regarding this revision.

Comment 1

Methods

Were all values normally distributed?

Response

Thank you for your reminding. Based on this comment, we did a series of revisions as below.

  1. We used Shapiro-Wilk test to examine thenormality of continuous variables in this study. We found that the scores of CES-D, APGAR Index, stress-coping orientation on the COPE, and behavioral and ADHD symptoms on the CBCL/6-18 were not normally distributed. Therefore, we transformed these continuous variables into dichotomous variables by using the correspondent cutoffs for each instrument. We revised the contents of Methods section regarding the introduction of the measures as below.

A CES-D cutoff of ≥21 was used to identify individuals with significant depressive symptom [52].” Please refer to line 161-162.

Because that the data was non-normally distributed, we defined total score above and below the 75th percentile of all participants in this study as high and low orientations of the stress-coping strategies, respectively. Accordingly, the cutoffs of problem-focused coping, emotion-focused coping, and less effective coping were 71, 57, and 27, respectively. Please refer to line 171-175.

Because that the data was non-normally distributed, we defined total score above and below the 25th percentile of all participants as perceiving high and low family support, respectively. Accordingly, the cutoff of the APGAR Index was 14. Please refer to line 181-184.

Because the CBCL/6-18 defined those with a T score higher than 70 (two standard deviation of the mean) to have severe symptoms [56-58], we used a cutoff of >70 to identify children with severe symptoms. Please refer to line 191-193.

  1. Because the dependent variable changed from the continuous variable (total CES-D score) to dichotomous one (significant depressive symptoms based on the cut-off of 21 on the CES-D), we reanalyzed the data using logistic regression analysis instead of regression analysis. We rewrote the content of “2.3. Statistical analysis” as below.

The predictive effects of caregivers’ demographic characteristics, significant depressive symptom at baseline, perceived low family support, and high orientation of stress-coping strategies, children’s demographic characteristics, and severe behavioral problems and ADHD symptoms (independent variables) on caregivers’ significant depressive symptoms at follow-up (dependent variable) were firstly examined by using univariate logistic regression analysis. Predictors that significantly increased the risk of significant depressive symptoms in univariate logistic regression further entered into multivariate logistic regression analysis. Please refer to line 201-208.

  1. We also revised the contents of Results and Discussion section based on the new results of statistical analysis. Compared with the results of the original analysis, two different results were found in the new analysis. First, perceived family support became significant but marriage status became nonsignificant when we examined their predictions for the risk of significant depressive symptoms in multivariate logistic regression analysis. Second, adopting problem-solving strategies significantly decreased the risk of significant depressive symptoms. We revised the Results section as below. Please refer to line 259-264.

“The results of Model 1 demonstrated that caregivers’ low family support and high orientation of less effective coping, and children’s high internalizing problems and ADHD symptoms increased the risk of caregivers’ significant depressive symptoms at follow-up, whereas caregivers’ high orientation of problem-focused coping decreased the risk of caregivers’ significant depressive symptoms at follow-up.”

Comment 2

What kind of stepwise selection was used (backward, forward?)

Response

As described in the response to Comment 1, we reanalyzed the data using logistic regression analysis instead of regression analysis. We rewrote the content of “2.3. Statistical analysis.” Please refer to line 201-208.

Comment 3

How was education level determined?

Response

The present study determined caregivers’ total length of formal education by calculating their total years of receiving formal education. We added the explanation into “2.2.5. Demographic characteristics.” Please refer to line 196-197.

We also determined caregivers’ total length of education by calculating their total years of receiving formal education.

Comment 4

What statistical software was used?

Response

Thank you for your reminding. We added the statistical software as below into the revised manuscript. Please refer to line 199.

“Data analysis was performed using SPSS 24.0 (SPSS, Chicago, IL, USA).”

Comment 5

Table 1. Was the highest value for years of education truly 28 years?

Response

We rechecked the data and confirmed the value of “28 years” is true. It might include the study in master and PhD programs.

Comment 6

Table 1. Please add n and % of common ADHD comorbidities

Response

The present study did not survey psychiatric comorbidity in children with ADHD. Instead, we added the number and percentage of ADHD children with severe internalizing and externalizing symptoms into Table 1.

Comment 7

Table 2. Please report 95% CI for comparison of metric values

Response

We reported odds ratio (OR) and its 95% CI based on the results of logistic regression in Tables 2 and 3.

Comment 8

Table 3. Please provide the reference category for the independent categorical variables (e.g. marriage status).

Response

Thank you for your reminding. We added the reference category into the footnote of Tables 2 and 3. Please refer to line 250-258 and 273-276.

Comment 9

Table 3. What was the Durbin Watson statistic?

Response

We changed the statistical analysis from multiple regression analysis to logistic regression analysis; therefore, the Durbin Watson statistic was not reported in the revised manuscript.

Comment 10

Please also describe the cohort in more detail. Instead of just reporting the mean values, you might consider to report number of people with different degrees of depression: The CES-D also provides cutoff scores (e.g., 16 or greater) that aid in identifying individuals with depression.

Response

Thank you for your suggestion. In the revised manuscript we used the cutoff of 21 on the CES-D based on the study of Zhang et al. (2015) to identify participants with significant depressive symptoms. We added the proportion of caregivers with significant depressive symptoms as below into Table 1 and Results section as below. Please refer to line 226-231.

“In total, 83 (21.7%) and 89 (23.3%) caregivers reported significant depressive symptoms at baseline and at-follow-up, respectively. Of the 83 caregivers with significant depressive symptoms at baseline, 55 (66.3%) still had significant depressive symptoms at follow-up. Of the 299 caregivers without significant depressive symptoms at baseline, 34 (11.4%) turned to report significant depressive symptoms at follow-up.”

Comment 11

Please report use of antidepressive medication in the participants.

Response

We did not determine the history of depressive disorders and treatment in caregivers with significant depressive symptoms. We listed it as one of limitations in this study as below. Please refer to line 383-386.

“… we did not determine the history of depressive disorders and treatment in caregivers with significant depressive symptoms. A long history of depressive disorders and lack of treatment may have deleterious effects on caregiver-child interaction and the recovery of caregivers’ depressive symptoms.”

Comment 12

Please detail the ADHD symptoms.

Response

We added the proportions of children with severe internalizing, externalizing, and ADHD symptoms Based on the scores of the CBCL/6-18 into Table 1 and Results section as below. Please refer to line 231-233.

“Based on the scores of the CBCL/6-18, the proportions of children with severe internalizing, externalizing, and ADHD symptoms were 13.1%, 17.3%, and 21.7%, respectively.”

Comment 13

Consider to report how your cohort is compared to other cohorts; e.g., normative data are available for the Child Behavior Checklist For Ages.

Response

We compared the proportions of children severe internalizing, externalizing, and ADHD symptoms in the present study with those in the normative data of the CBCL/6-18 as below. Please refer to line 342-347.

“The proportions of children with severe internalizing, externalizing, and ADHD symptoms determined by the cutoff of a T score > 70 (2 SD) on the CBCL/6-18 were 13.1%, 17.3%, and 21.7%, respectively in this study. Because that only about 2.3% of children in the general population have a T score > 70 on the CBCL/6-18 [56-58], the proportions of children with severe internalizing, externalizing, and ADHD symptoms in this study were high.”

Comment 14

Instead of using significant variables listed in Table 2 for stepwise multiple regression analysis, you can use all variables with backward selection and AIC criterion.

Response

In the revised manuscript we changed the statistical method into univariate and multivariate logistic regression analysis. Please refer to line 201-207.

Comment 15

Was there a correlation between different coping styles and depression?

Response

Yes, the results of multivariate logistic regression analysis revealed that adopting less effective coping strategies increased the risk of significant depressive symptoms, whereas adopting problem-solving coping strategies decreased the risk of significant depressive symptoms. In addition to the original contents describing the relationship between adopting less effective coping and depression, we added discussion regard the relationship between adopting problem-solving coping and depression as below. Please refer to line 333-337.

Otherwise, research found that adoption of problem-solving coping strategies such as active coping and planning can mobilize people to analyze stressful situations and attempt to resolve them [67]. Caregivers adopting of problem-solving coping strategies may not only have the increased opportunity to successfully resolve stressful situations but also get positive rewards to improve self-esteem.”

Comment 16

Discussion

"21.7% of caregivers were 223 identified to have significant depression" this belongs to the results.

Response

We moved this sentence to Results section. Please refer to line 226.

Comment 17

Coping: Please also discuss your findings in terms of the categorization into active and passive coping.

Response

Thank you for your suggestion. As described in response to Comment 15, we added discussion regard the relationship between adopting problem-solving coping and depression as below, in addition to the original contents describing the relationship between adopting less effective coping and depression. Please refer to line 333-337.

Otherwise, research found that adoption of problem-solving coping strategies such as active coping and planning can mobilize people to analyze stressful situations and attempt to resolve them [67]. Caregivers adopting of problem-solving coping strategies may not only have the increased opportunity to successfully resolve stressful situations but also get positive rewards to improve self-esteem.”

Comment 18

The main problem in this paper is the missing information about relevant cofactors: duration of depression (existence of clinical significant depression according to DSM criteria), changes of therapy of ADHD symptoms, changes of ADHD symptoms during study period.

Response

Thank you for your comment. We added them into the limitations of this study as below. Please refer to line 382-386.

“We also did not collect the data of treatment for children’s ADHD at follow-up. Moreover, we did not determine the history of depressive disorders and treatment in caregivers with significant depressive symptoms. A long history of depressive disorders and lack of treatment may have deleterious effects on caregiver-child interaction and the recovery of caregivers’ depressive symptoms.”

Comment 19

In general the term "depression" is somehow misleading, because the CES-D asks symptoms associated with depression, but it is not a instrument to diagnose depression.

Response

Thank you for your comment. We agree that the CES-D is an instrument for measuring the severity of depressive symptoms but not for making a diagnosis. In the revised manuscript we used the cutoff of 21 on the CES-D to identify participants with significant depressive symptoms. Therefore, we replaced “depression” unto “significant depressive symptoms“ in the title and thorough the revised manuscript.

Comment 20

The paper would also profit from more detailed descriptive statistics, e.g.  the number of different depression symptoms according to the CES-D, different ADHD symptoms.

Response

We added the number of caregivers with significant depressive symptoms on the CES-D into Table 1 and Results section (Please refer to line 226-231). We also added the number of children with severe ADHD symptoms on the CBCL/6-18 into Table 1.

Reviewer 3 Report

The study is well thought out, similar to that of Oman:  https://www.tandfonline.com/doi/full/10.1080/03004430.2017.1394850

There are limitations (receiving treatment) to be considered as a description of the sample. Some variables poorly explained.

the authors have similar publications: https://www.mdpi.com/1660-4601/18/14/7532 

Author Response

Thank you for your comments. As discussed below, we have revised our manuscript based on your comments. Please let us know if we need to provide anything else regarding this revision.

Comment 1

The study is well thought out, similar to that of Oman:  https://www.tandfonline.com/doi/full/10.1080/03004430.2017.1394850

Response

Thank you for your reminding. We cited this study into the revised manuscript as Reference 4 (Al-Balushi , N.; Al Shekaili, M.; Al-Alawi, M.; Al-Balushi, M.; Panchatcharam, S.M.; Al-Adawi, S. Prevalence and predictors of depressive symptoms among caregivers of children with attention-deficit/hyperactivity disorder attending a tertiary care facility: a cross-sectional analytical study from Muscat, Oman. Early Child Dev. Care. 2019, 189, 1515-1524. doi 10.1080/03004430.2017.1394850.). Please refer to line 44, 62 and 63.

Comment 2

There are limitations (receiving treatment) to be considered as a description of the sample.

Response

We agreed that lack of information regarding treatment is a limitation of this study. In the original manuscript we had listed lack of data for treatment of caregivers’ depression as one of limitations (please refer to the second limitation in the section “4.5. Strengths and limitations”). We further added lack of information on treatment for children’s ADHD as the limitation of this study as below. We also added the lack of the data for history of depressive disorders in caregivers as the limitation of this study as below.

We also did not collect the data of treatment for children’s ADHD at follow-up. Please refer to line 382.

“…we did not determine the history of depressive disorders and treatment in caregivers with significant depressive symptoms. A long history of depressive disorders and lack of treatment may have deleterious effects on caregiver-child interaction and the recovery of caregivers’ depressive symptoms.” Please refer to line 383-386.

Comment 3

Some variables poorly explained.

Response

Thank you for your comment. We found that we did not introduce the predictions of depressive symptoms and marriage status at baseline for depressive symptoms at follow-up in caregivers of children with ADHD in Introduction section. We added two paragraphs into Introduction section as below to describe them.

“1.5. Prediction for significant depressive symptoms in caregivers based on perceived family support and marriage status

Research has found that parents of children with ADHD are more likely to divorce compared with those of children without ADHD [38], as well as that divorce can cause health-related immune alterations and increase the risk of depressive problems [39]. Whether a marriage status of separation or divorce can predict caregivers’ depressive problems warrants further study. Please refer to line 104-109.

“1.6.Persistent depressive symptoms in caregivers of children with ADHD

Research has found that caregivers of children with ADHD had heavy care burdens [40-42] and high parenting stress [43,44]. Caregivers might also experience public stigma toward ADHD [45] and have affiliate stigma toward themselves, which may further increases their psychological distress [46] and the risk of depressive problems [47]. It is reasonable to hypothesize that caregivers’ depressive symptoms may persist or even worsen during the growth stage of children with ADHD; therefore, significant depressive symptoms at baseline may predict the non-remitted depressive symptoms at follow-up. However, whether the predictive effect exists warrants further study.” Please refer to line 110-118.

Comment 4

the authors have similar publications: https://www.mdpi.com/1660-4601/18/14/7532 

Response

In the previous publication we aimed to examine the prediction of affiliate stigma for caregivers’ and children’s emotional and behavioral problems (Chang, C.C.; Chen, Y.M.; Hsiao, R.C.; Chou, W.J.; Yen, C.F. Did affiliate stigma predict affective and behavioral outcomes in caregivers and their children with attention-deficit/hyperactivity disorder? Int. J. Environ. Res. Public Health. 2021, 18, 7532). We found that caregivers’ affiliate stigma at baseline positively predicted children’s affective and somatic problems 1 year later after controlling for caregivers’ depression and children’s behavioral problems at baseline. The aim of the previous publication differed from those of the present one. We added the result of the previous publication as below into the revised manuscript. Please refer to line 120-124.

“This prospective study followed the participants of the Study on Affiliate Stigma in Caregivers of Children with ADHD [47] 1 year later. In the previous analysis, we found that caregivers’ affiliate stigma at baseline positively predicted children’s affective and somatic problems 1 year later after controlling for caregivers’ depression and children’s behavioral problems at baseline [48].

Round 2

Reviewer 2 Report

Thank you for the revision. Most comments were adressed. Some points remain.

  • The term "significant depressive symptoms" is misleading. Delete the "significant".
  • As authors checked (they should do these basic statistical methods before submitting a paper) most values were not normally distributed. However, it is not necessary and not reasonable to dichotomize metric values only because they are not normally distributed. A better way would be to use non-parametric methods or to transform values if necessary (e.g., Johnson transformation). They can also use common linear regression as long as other prerequisites are fullfilled (no multicolinaraity, independence of residuals etc.).  By dichotomizing metric values they loose a lot of information and power. At least, I recommend to add the results using (non-parametric) linear regression methods (e.g. as supplement). This would help to support their findings. The should also discuss different findings between logistic and linear regression.
  • Please report godness of fit measures for logistic regression (Nagelkerke R2 etc).

Author Response

Thank you for your comments. As discussed below, we have revised our manuscript based on your comments. Please let us know if we need to provide anything else regarding this revision.

Comment 1

The term "significant depressive symptoms" is misleading. Delete the "significant".

Response

We deleted the word “significant” from the revised manuscript.

Comment 2

As authors checked (they should do these basic statistical methods before submitting a paper) most values were not normally distributed. However, it is not necessary and not reasonable to dichotomize metric values only because they are not normally distributed. A better way would be to use non-parametric methods or to transform values if necessary (e.g., Johnson transformation). They can also use common linear regression as long as other prerequisites are fullfilled (no multicolinaraity, independence of residuals etc.).  By dichotomizing metric values they loose a lot of information and power. At least, I recommend to add the results using (non-parametric) linear regression methods (e.g. as supplement). This would help to support their findings. The should also discuss different findings between logistic and linear regression. Please report godness of fit measures for logistic regression (Nagelkerke R2 etc).

Response

Thank you for your comment. We agree that dichotomizing metric values may lose a lot of information and power. We consulted a statistical expert and re-examined our data. According to Kim (2013), the absolute vales of skewness < 2 and kurtosis < 7 indicate normal distribution if the sample size was larger than 300. We found that the continuous variables of this study were normally distributed. Therefore, we re-analyzed the data using univariate and backward selection of multivariate linear regression analysis. Compared with the original results of logistic regression analysis, the results of multivariate regression analysis found that caregivers’ marriage status of separation or divorce at baseline became significantly associated with depressive symptoms at follow-up. Moreover, the orientation of emotion-focus coping became significantly associated with depressive symptoms at follow-up, whereas the significant association between the orientation of problem-solving coping and depressive symptoms at follow-up became nonsignificant. We revised our manuscript as below.

  1. Reference 58:

Kim, H.Y. Statistical notes for clinical researchers: assessing normal distribution (2) using skewness and kurtosis. Restor. Dent. Endod. 2013, 38, 52-54. doi: 10.5395/rde.2013.38.1.52.

  1. Abstract: Please refer to line 30-35.

Caregivers’ marriage status of separation or divorce, orientation of less effective coping and depressive symptoms, and children’s internalizing problems and ADHD symptoms at baseline were positively associated with caregivers’ depressive symptoms at follow-up, whereas caregivers’ perceived family support and orientation of emotion-focused coping at baseline were negatively associated with depressive symptoms at follow-up.

  1. Methods: 2.3. Statistical analysis

“We used skewness and kurtosis to examine whether the continuous variables normally distributed. According to Kim [58], the absolute vales of skewness < 2 and kurtosis < 7 indicate normal distribution if the sample size was larger than 300.” Please refer to line 189-191.

“… univariate linear regression analysisbackward selection of multivariate linear regression analysis. Please refer to line 196 and 198.

“The Durbin Watson statistic was used test the autocorrelation in the residuals.” Please refer to line 202-203.

  1. Results

The absolute vales of skewness of the continuous variables ranged from 0.101 to 1.034; the absolute vales of kurtosis ranged from 0.014 to 1.170. The values of skewness and kurtosis indicated that the continuous variables were normally distributed. Please refer to line 222-225.

Table 2 presents the results of univariate linear regression examining the associations of caregivers’ and children’s factors at baseline with caregivers’ depressive symptoms at follow-up. Regarding caregivers’ factors, the results indicated that compared with the caregivers with the marriage status of being married or cohabited at baseline, the caregivers with the marriage status of separation or divorce had greater depressive symptoms at follow-up. Depressive symptoms at baseline and the orientation of less effective coping were positively associated with depressive symptoms at follow-up, whereas perceived family support, the orientations of problem-focused coping and emotion-focused coping were negatively associated with depressive symptoms at follow-up. Regarding children’s factors, internalizing and externalizing problems and ADHD symptoms were positively associated with caregivers’ depressive symptoms at follow-up. Please refer to line 229-239.

Factors that were significantly associated with caregivers’ depressive symptoms in univariate linear regression analysis were selected into backward selection of multivariate linear regression analysis to examine their association with caregivers’ depressive symptoms at follow-up (Table 3). The results of Model 1 demonstrated that caregivers’ marriage status of separation or divorce and orientation of less effective coping and children’s internalizing problems and ADHD symptoms at baseline were positively associated with caregivers’ depressive symptoms at follow-up, whereas caregivers’ perceived family support and orientation of emotion-focused coping at baseline were negatively associated with depressive symptoms at follow-up. Caregivers’ depressive symptoms at baseline was further selected into multivariate linear regression analysis. The result of Model 2 indicated that depressive symptoms at baseline was significantly associated with depressive symptoms at follow-up in caregivers. Multicollinearity among variables was not detected because the tolerance and VIF values ranged from 0.758 to 0.988 and from 1.012 to 1.319, respectively, and the condition index value was 26.976. The value of the Durbin Watson statistic was 1.974, indicating a weak autocorrelation in the residuals. Please refer to line 244-258.

  1. Discussion

The present study also found that caregivers’ marriage status of separation or divorce at baseline predicted depressive symptoms 1 year later. Divorce can not only cause health-related immune alterations and increase the risk of depression [39] but also result in a lack of a coadjutant to manage the behaviors of children with ADHD, thus increasing caregivers’ parental distress. A previous study on the parents of children with psychopathology found that parents who did not live together had a higher level of depression than did those who lived together [17]. The results of this study indicated that enhancing family support, especially for caregivers with the marriage status of separation or divorce is important to improve caregivers’ mood regulation. Please refer to line 287-296.

This study demonstrated that caregivers’ emotion-focused coping at baseline was negatively associated with depressive symptoms at follow-up. Parents may experience multiple sources of stress when taking care of children with ADHD such as managing their children’s behaviors, supervising the completion of their children’s daily routines, maintaining relationships among family members, and craving social support [35]. Emotion-focused coping such as seeking socioemotional support, positive reinterpretation, acceptance of stressors, and perceived growth in the process of managing stress may diminish the negative emotional consequences of stressful events [22,23].Please refer to line 222-225.

  1. Acknowledgement: The authors gratefully acknowledge Dr. Chung-Ying Linfor his assistance in statistical analysis.” Please refer to line 384-385.